# Cardamonin Exerts Antitumor Effect on Human Hepatocellular Carcinoma Xenografts in Athymic Nude Mice through Inhibiting NF-κβ Pathway

**DOI:** 10.3390/biomedicines8120586

**Published:** 2020-12-09

**Authors:** Nassrin Badroon, Nazia Abdul Majid, Fouad Saleih R. Al-Suede, Mansoureh Nazari V., Nelli Giribabu, Amin Malik Shah Abdul Majid, Eltayeb E. M. Eid, Mohammed Abdullah Alshawsh

**Affiliations:** 1Institute of Biological Sciences, Faculty of Science, University of Malaya, Kuala Lumpur 50603, Malaysia; nabtqm@hotmail.com; 2Department of Pharmacology, Faculty of Medicine, University of Malaya, Kuala Lumpur 50603, Malaysia; 3EMAN Biodiscoveries Sdn. Bhd., Kedah Halal Park, Kawasan Perindustrian Sungai Petani, Sungai Petani 08000, Malaysia; fsr2009@yahoo.com (F.S.R.A.-S.); nazarimansoure@gmail.com (M.N.V.); 4Department of Physiology, Faculty of Medicine, University of Malaya, Kuala Lumpur 50603, Malaysia; nelli.giribabu@um.edu.my; 5Eman Biodiscoveries Sydney Bhd., and ACRF Department of Cancer Biology and Therapeutics, The John Curtin School of Medical Research, Australian National University, 131 Garran Road, 2601 Acton, Australia; aminmalikshah@gmail.com; 6Department of Pharmaceutical Chemistry and Pharmacognosy, Unaizah College of Pharmacy, Qassim University, Unaizah 51911, Saudi Arabia; eem.eid@qu.edu.sa

**Keywords:** cardamonin, hepatocellular carcinoma, anti-tumorigenesis, molecular docking, NF-κB pathway

## Abstract

Cardamonin (CADMN) exerts an in vitro antiproliferative and apoptotic actions against human hepatocellular carcinoma cells (HepG2). This study aimed to investigate the in vivo anti-tumorigenic action of CADMN against human hepatocellular carcinoma xenografts in an athymic nude mice, as well as to study the molecular docking and safety profile of this compound. Acute toxicity study demonstrated that CADMN is safe and well-tolerated up to 2000 mg/kg in ICR mice. Oral administration of 50 mg/kg/day of CADMN in xenografted nude mice showed a significant suppression in tumor growth as compared to untreated control group without pronounced toxic signs. Immunohistochemistry assay showed downregulation of proliferative proteins such as PCNA and Ki-67 in treated groups as compared to untreated control. Additionally, immunofluorescence analysis showed a significant downregulation in anti-apoptotic Bcl-2 protein, whereas pre-apoptotic Bax protein was significantly upregulated in nude mice treated with 25 and 50 mg/kg CADMN as compared to untreated mice. The findings also exhibited down-regulation of NF-κB-p65, and Ikkβ proteins, indicating that CADMN deactivated NF-κB pathway. The molecular docking studies demonstrated that CADMN exhibits good docking performance and binding affinities with various apoptosis and proliferation targets in hepatocellular cancer cells. In conclusion, CADMN could be a potential anticancer candidate against hepatocellular carcinoma. Other pharmacokinetics and pharmacodynamics properties, however, need to be further investigated in depth.

## 1. Introduction

Although the geographical mapping of hepatocellular carcinoma (HCC) indicates that the incidence and mortality rate of HCC has changed over time, HCC is still the fourth most common cause of cancer deaths worldwide [1]. In addition, the incidence rate of HCC was reported to vary considerably across the world, with the highest incidence observed in East Asia [2]. On the other hand, several problems have arisen in the treatment of HCC, including drug resistance and high recurrence rate [3]. Furthermore, the chemotherapeutic agents are also associated with undesirable effects such as toxicity, fever, nausea, infection, abdominal pain, tiredness, and weakness. The absence of curative treatment for HCC encouraged scientists to conduct an intensive pharmacological studies to find alternative anticancer agents with better safety and effectiveness profile to treat HCC [4,5]. Cardamonin (CADMN) is a chalconoid that has been extracted from cardamom spice, *Alpinia gagnepainii*, *Boesenbergia rotunda*, *Piper hispidum* and others medicinal plants [6]. The cytotoxic activities of CADMN have been reported against several cancer cell lines including U266 (myeloma), A549 (lung), MDA-MB-231 (breast), DU145 (prostate), MCF-7 (breast), and SGC7901 (gastric) [7,8,9,10,11]. In animals, CADMN also demonstrated inhibitory effects on tumor growth in mice through modulation of different pathways such as Wnt/β-catenin, mTOR, and NF-κB signaling pathways [10,12,13,14]. In our earlier published in vitro study, CADMN has been reported to exert anti-proliferative and apoptotic action in hepatocellular carcinoma cells (HepG2) through enhancing the ROS accumulation, which lead to an inhibition of NF-κB translocation [15]. However, the anti-tumorigenesis activity of CADMN against HCC has yet to be investigated in animal models. The current study aimed to evaluate the acute toxicity and anti-tumorigenesis properties of CADMN against HCC xenografts in nude mice as well as to study the molecular docking of CADMN on different related targets.

## 2. Experimental Section

### 2.1. Chemicals and Compounds

Cardamonin with molecular weight 270.28 g/mol and purity > 98% was purchased from Sigma Aldrich, MO, USA. For in vivo experiments, cardamonin was dissolved in 10% Tween 20. The 5-Fluorouracil (5-FU) was purchased from MP Biomedical, lllkirch, France. All other chemicals were purchased from Fisher and Sigma with analytical grade.

### 2.2. Cell Line

HepG2 human HCC cells (derived from the liver tissue of a 15-year-old American adolescent boy of European ancestry with a well-differentiated hepatocellular carcinoma) was purchased from American Type Culture Collection (ATCC), Manassas, VA, USA.

### 2.3. Animals

For acute toxicity, twelve healthy female ICR mice (8–12 weeks; 22–28 g of body weight) were obtained from Animal Experimental Unit, Eman Research Limited. For anticancer study, 24 healthy male athymic nude mice (4–6 weeks, 20–22 g of body weight) were purchased from animal experimental unit, Eman Research Limited.

Acute toxicity study and in vivo xenografts in an athymic nude mice experiments were conducted after obtaining the ethical approval (ethic.112069A182420120; approved: 02/01/2020) issued by Animal Care Board, Eman Research Animal Ethics Committee, Malaysia. All mice were received humane care according to the conditions summarized in the Guide for the Care and Use of Laboratory Animals [16].

### 2.4. Acute Toxicity Study

The lethal dose (LD_50_) of CADMN was investigated according to OECD guideline test No. 423 for acute oral toxicity [17]. Mice (3 per cage) were let to acclimatize for one week under standard laboratory conditions of 22–24 °C, 12/12-h light/dark conditions and 50–60% humidity. The 12 mice were randomly distributed into four groups (n = 3 mice/group), and three groups were orally administered with 50, 300, and 2000 mg/kg BW of CADMN via oral gavage, respectively, while the fourth group served as a negative control (untreated). Starting dose was 300 mg/kg BW (0.2 mL) administered orally and the compound was dissolved in 10% tween 20 [17]. Mice were observed for toxicity signs and behavioral changes during the first one hour after dosing and periodically for the next 24 h, and then daily for 14 days. After 14 days of the treatment, all mice were euthanized, and liver and kidney were harvested for histopathology. Body weight of mice was recorded at baseline (the first day of enrolling mice into the treatment) and later every three days.

### 2.5. In Vivo Xenografts in Athymic Nude Mice Model

#### 2.5.1. Experimental Design

HepG2 cells were cultured in Eagle’s Minimum Essential Medium (EMEM), which were supplemented with 1% penicillin/streptomycin, 10% fetal bovine serum (FBS) and maintained in a 37 °C incubator with 5% CO_2_. HepG2 liver cancer cells (2 × 10^6^ in 200 µL media) mixed with Matrigel were injected subcutaneously into the right flank of the nude mice [18]. A total of 24 nude mice were randomly distributed into 4 groups (6 mice/group) including untreated tumor control group (given 10% tween 20, oral gavage daily), low dose CADMN-treated tumor group (given 25 mg/kg CADMN in 10% Tween 20 oral gavage daily), high dose CADMN-treated tumor group (given 50 mg/kg CADMN in 10% tween 20 oral gavage daily) and 5-floururacil- treated tumor group as a positive control (given 30 mg/kg 5-FU, intraperitoneal 3 times/week) [19]. The treatment was initiated immediately once the tumor volume reaches 100 mm^3^ utilizing oral gavage according to the respective treatments for 24 days.

#### 2.5.2. Tumor Growth Monitoring

HepG2 tumor growth was observed by measuring the size of the tumor using a digital caliper weekly and at the sacrificing date. The tumor volume was calculated as follows:(1)$$V=\frac{a\times b^{2}}{2}$$
where a = the long diameter, b = short diameter, and V = volume (mm^3^) of tumors in the mice.

The inhibition of tumor in each group was calculated with the formula [20]:(2)$$\mathrm{Inhibition}\;(\%)\;=\;1\;-\;\frac{\mathrm{mean}\;\mathrm{tumor}\;\mathrm{volume}\;\mathrm{on}\;\mathrm{scarificing}\;\mathrm{date}\;-\;\mathrm{mean}\;\mathrm{tumor}\;\mathrm{volume}\;\mathrm{at}\;\mathrm{baseline}}{\mathrm{mean}\;\mathrm{tumor}\;\mathrm{volume}\;\mathrm{of}\;\mathrm{control}\;\mathrm{on}\;\mathrm{scarificing}\;\mathrm{date}\;-\;\mathrm{mean}\;\mathrm{tumor}\;\mathrm{volume}\;\mathrm{of}\;\mathrm{control}\;\mathrm{at}\;\mathrm{baseline}}\;\times \;100$$

After 24 days of treatment, the animals were fasted overnight. The mice were anesthetized using ketamine 100 mg/kg and xylazine 10 mg/kg as a single intraperitoneal (IP) injection and blood samples were collected via cardiac puncture for biochemical analysis. Then cervical dislocation was performed to confirm the death. The tumor was harvested, washed with cold phosphate buffer saline (PBS), and the tumor weight was recorded. The tumor was fixed in 10% buffered formaldehyde solution for histological examination and immunohistochemistry [3,21].

#### 2.5.3. Biochemical Analysis

Blood samples from all nude mice were collected via cardiac puncture under anesthesia. The serum biochemistry parameters analysis was performed for liver and kidney function including AST (aspartate aminotransferase), ALT (alanine aminotransferase), ALP (alkaline phosphatase), total bilirubin, creatinine, and urea. All biochemical parameters were analyzed spectrophotometrically by standard automated techniques following the manufacturer instructions.

#### 2.5.4. Histological Analysis of Tumor

Gross post-mortem examinations were performed on all terminated animals. The tumor was fixed in 10% buffered formalin then processed in the paraffin wax. Approximately 4 µm sections from blocks were stained with H and E (hematoxylin-eosin) stain and examined under microscope [3].

#### 2.5.5. Immunohistochemistry

Immunohistochemical analyses were performed using the commercial kit of streptavidin-biotin and peroxidase protocol according to the manufacturer’s structure (Dako, Carpinteria, CA, USA) to detect Ki-67 (cat # ab16667) and proliferating cell nuclear antigen (PCNA/cat No. E-AB-32521) proteins. Tissue parts were fixed in 10% buffered formalin and then dehydrated with a series grade of ethanol and cut at 4 μm sections. These sections were deparaffinised and rehydrated in graded series of ethanol. After that, the sections were incubated in a microwave for antigen retrieval using (10 mM sodium citrate buffer). Then, the processed tissues were rinsed with PBS and endogenous peroxidase, and were blocked with 0.3% H_2_O_2_ for 30 min. The sample of the tissue was washed gently by wash buffer, then biotinylated primary antibodies including Ki-67 (1:100) and PCNA (1:200) were added and incubated for 15 min followed by washing with PBS. After that, the samples were placed in buffer bath in a humid chamber. Then, for 20 min, streptavidin-peroxidase conjugated to horseradish peroxidase was added and incubated followed by gentile rinsing. After that, the samples were incubated for 5 min with diaminobenzidine (DAB)-substrate chromogen followed by washing with PBS and immersed in hematoxylin for 5 s. The samples were washed then dipped for 10 times in 0.037 M/l of ammonia. The same steps were applied to the negative control sections but without adding the primary antibodies. A bath of deionized water was used to wash the samples for 2–5 min. Brown staining appeared under the optical microscope reflecting the positive antigens [3].

#### 2.5.6. Immunofluorescence Staining Assay

Immunofluorescence staining analysis was performed to detect Bax (E-AB-33819), Bcl-2 (E-AB-60012), IKKβ (cat# ab124957), NF-κB-p65 (cat# 6956S), and p-NF-κB-p65 (cat#3033S) proteins. Tumor tissues were fixed in 10% buffered formalin, then dehydrated with a series grade of ethanol and cut at 4 μm sections. These sections were deparaffinised with xylenes and rehydrated in graded series of alcohol. After that, heat treatment was performed on the sections for antigen unmasking using 10 mM sodium citrate buffer. Then, the processed tissues were rinsed with deionized/Milli-Q water three times for 2 min. Specimens were incubated with 5–10% normal blocking serum (normal sera for immunohistochemistry) in PBS solution for 1 h to suppress the non-specific binding of IgG. Then, specimens were incubated with primary antibody including NF-κB-p65 (L8F6) mouse mAb (Cat#6956S) and p-NF-κB-p65 (S536) (93H1) Rabbit mAb (Cat#3033S) for overnight followed by washing with three changes of PBS for 5 min each. The same procedures were applied to Bax polyclonal Ab (E-AB-33819), Bcl-2 polyclonal Ab (E-AB-60012) and IKKβ mAb (cat# ab124957). After that, the specimens were incubated for 45 min with secondary antibody. Specimens were incubated with streptavidin-fluorescein for 15 min in a dark chamber. Then, mount coverslip with UltraCruzTM mounting media (DAPI). The same steps were applied to the negative control sections but without adding the primary antibodies. The samples were observed using a fluorescence microscope. The intensity of fluorescence was quantified via ImageJ software.

### 2.6. Molecular Docking

#### 2.6.1. Proteins Preparation

The three dimensional crystal structure of anticancer targets Bax BH3 with PDB ID: 4BD6, Bcl-2 with PDB ID: 5WHI, cytochrome C with PDB ID: 4CDA, caspase 7 with PDB ID: 4LSZ, caspase 8 with PDB ID: 3KJN, caspase 9 with PDB ID: 1JXQ, caspase 3 with PDB ID: 3KJF, PCNA with PDB ID: 3WGW, NF-κB with PDB ID: 2DBF, and NF-κB-p65 with PDB ID: 1MY7 were selected for this study and downloaded from Protein Data Bank (www.rcsb.org/pdb) (Appendix A) [22]. The complexes bound to the receptor molecule, non-essential water molecules and heteroatoms were removed and ultimately hydrogen atoms were added to the target receptor molecule using Argus Lab.

#### 2.6.2. Software

Python 3.9.0 language was downloaded from www.python.org, Molecular Graphics laboratory (MGL) tools 1.5.7 was downloaded from http://mgltools.scripps.edu and AutoDock4.2 was downloaded from http://autodock.scripps.edu, BIOVIA Draw and Discovery studio visualizer 2017 were downloaded from http://accelrys.com, and Chem3D was downloaded from https://acms.ucsd.edu.

#### 2.6.3. Ligand Preparation

Cardamonin was already available with known structure from crystallography and was downloaded from PubChem database in sdf format and converted to PDB format using Pymol and further used for docking studies. The starting structures of the proteins were prepared using AutoDock tools. The water molecule was deleted, and polar hydrogen and Kollman charges were added to the protein starting structure. Grid box was set with the size of 126 × 126 × 126 Å with the grid spacing of 0.375 Å at the binding site. The starting structure for ligand was constructed using BIOVIA Draw. 5-FU was selected as positive control. Their structures were provided from the PubChem database. Gasteiger charges were assigned into optimized ligand using AutoDock tools. In total, 100 docking runs were conducted with a mutation rate of 0.02 and crossover rate of 0.8. The population size was set to use 250 randomly placed individuals. Lamarckian Genetic algorithm was used as the searching algorithm with a translational step of 0.2 Å, a quaternion step of 5 Å and a torsion step of 5 Å. Most populated and lowest binding free energy [23].

### 2.7. Statistical Analysis

Data were presented as mean ± standard deviation. Statistical analysis was performed with GraphPad Prism, version 5 for windows and SPSS program for windows version 20 (SPSS Inc., IL, USA). Significance was performed using one-way analysis of variance (ANOVA), followed by post hoc Tukey’s for comparing treated groups to control group. *p* < 0.05 was considered as statistically significant.

## 3. Results

### 3.1. Acute Toxicity Study of CADMN

The results showed that all treated mice remained alive and there were no toxicity signs that appeared at different doses, even with the 2000 mg/kg dose. In addition, there were no significant changes in the body weight between treated and untreated mice (Figure 1a). Histopathology examination showed normal architecture of liver and kidney in mice treated with CADMN and untreated mice (Figure 1b). Thus, results indicate that the oral lethal dose (LD_50_) value of CADMN in female ICR mice is greater than 2000 mg/kg.

### 3.2. In Vivo Antitumor Activity of CADMN

#### 3.2.1. Tumor Volume, Tumor Weight, and Body Weight

Findings of tumors volume before and after the treatment with CADMN showed that it suppressed the tumor growth and CADMN-treated mice exhibited smaller tumor size as compared to untreated mice (Figure 2a).

The weekly tumor growth rate in CADMN-treated mice and 5-FU-treated group were slower as compared to untreated control group (Figure 2b). On the sacrificing date, the recorded data of the final tumor volume showed that 5-FU, 25 mg/kg and 50 mg/kg of CADMN groups were 474.7 ± 209.1 mm^3^, 776.7.7 ± 243.9 mm^3^ and 532.8 ± 285.6 mm^3^, respectively, which were significantly smaller as compared to the untreated group (1324 ± 493.9 mm^3^) (Figure 2c).

The results demonstrated the mean tumor weight of treated mice was significantly lower (1.6 ± 0.7 g for 25 mg/kg CADMN-treated group, 1.0 ± 0.7 g for 50 mg/kg CADMN-treated group and 0.9 ± 0.5 g for 30 mg/kg 5-FU-treated group) as compared to the untreated group (2.9 ± 1.0 g) (Figure 2d). As shown in (Figure 2e), there was no significant body weight loss was observed in all CADMN-treated groups throughout the treatment period. Administration of 25 mg/kg and 50 mg/kg of CADMN and 30 mg/kg of 5-FU resulted in an inhibitory rates (IR) of tumor volume by 45.4%, 65.2% and 70%, respectively. As illustrated in (Figure 2f), CADMN showed a dose-dependent inhibition of tumor growth in mice-treated groups.

#### 3.2.2. Blood Biochemistry Analysis

The biochemical serum parameters including kidney (creatinine and urea) and liver (AST, ALP, ALT and T-bilirubin) function markers were analyzed for both CADMN-treated and untreated nude mice (Table 1). The biochemical parameters analysis results showed that administration of CADMN in xenografts nude mice did not exhibit significant differences between untreated and CADMN-treated mice. Meanwhile, the results of 5-FU-treated mice showed significant increase in the liver function indicator (ALT) as compared to untreated group proving that 5-FU had toxicity to the liver.

#### 3.2.3. Tumor Histology Examination

A histological examination of tumor tissue samples using hematoxylin-eosin staining was performed. The sections of untreated group showed large solid tumor tissues and the tumor cells were very aligned and had bigger cell nucleus. Treated mice showed larger area of tumor necrotic characterized by cell shrinkage, fragmentation, and chromatin disappearance with a small island of tumor tissues (Figure 3). The results suggested that CADMN caused remarkable changes in tumor tissues.

#### 3.2.4. Immunohistochemistry and Immunofluorescence Staining Assay

The results of the immunohistochemistry staining showed that Ki-67 and PCNA proteins were strongly expressed in control group. However, the expression of both proteins decreased in CADMN-treated mice (Figure 4a), which indicate that CADMN exerts an anti-proliferative effect.

Immunofluorescence results showed that treatment with 50 mg/kg of CADMN resulted in a significantly lower expression levels of Bcl-2, p-NF-κB-p65 and Ikkβ when compared to untreated mice (Figure 4b–d). In contrast, treatment with 50 mg/kg of CADMN lead to a significant higher expression of Bax as compared to control.

### 3.3. Molecular Docking

The docked conformation of Bax BH3, Bcl-2, Cyt. C, caspase 7, caspase 8, caspase 9, PCNA, NF-κB, and NF-κB-p65 with active conformation of each ligand consists of CADMN and 5-FU clearly revealed that numerous potential interactions were present (Appendix A) and Table 2.

The interactions between Bax BH3 and ligands (CADMN and 5-FU) showed that free binding energy of CADMN is lower than 5-FU, which has been selected as positive control (Appendix A). Although 5-FU has shown two hydrogen bonds linked with ALA81 and ASP84, CADMN has shown three hydrogen bonds with PHE30, VAL50, and GLN52. In addition, CADMN has only one hydrogen binding more than 5-FU in interaction with Bax; however, the halogen binding that 5-FU possesses interacted with LYS123 of Bax, making the overall interactions weaker than the one in CADMN and Bax [24]. CADMN has shown three alkyl binding with Bax via LYS64, GLY29, and LYS57 [25]; however, CADMN has much lower free binding energy than 5-FU in interactions with Bax.

Appendix A shows the interaction of CADMN with Bcl-2 through three hydrogen bonds, while 5-FU has four hydrogen bonds with ALA109, ILE108, VAL62, and VAL64. Cardamonin also exhibited pi-alkyl binding with PRO24 and PRO30 and pi-sigma binding with GLN25. On the other hand, there are three hydrogen bonds with ASP15, GLN18 and SER28. Free binding energy of CADMN in interaction with Bcl-2 showed to be −6.27 kcal/mol, but this was −4.03 kcal/mol for 5-FU. In this interaction, comparison between CADMN and 5-FU showed more hydrogen bonds were present with 5-FU; however, the free binding energy of CADMN was still lower than positive control. This could be due to aromatic groups in the structure of CADMN involved in Pi-binding, which can stabilize the active pocket and cause lower binding energy as compared to the positive control [25].

As it is shown in Appendix A free binding energy of CADMN showed to be lower than 5-FU after interaction with caspase 3. Cardamonin has shown two hydrogen bonds in interaction with caspase 3, namely CYS163 and ARG64. On the other hand, 5-FU showed interactions with six hydrogen bonds via GLY122, CYS163, SER120, ARG64, GLN161, and ARG207. Moreover, CADMN showed Pi-pi stacking binding with TYR204. Even though 5-FU has shown more hydrogen binding with caspase 3 compared with CADMN, the free binding energy was still more. This is probably due to stabilization of active pocket by two aromatic groups that are available in the structure of CADMN. Free binding energy of CADMN has shown to be −7.16 kcal/mol, while it was −5.40 kcal/mol for 5-FU.

On the other hand, the interactions between caspase 7 and ligands (CADMN and 5-FU) showed that CADMN has two hydrogen bonds with LYS160 and GLU216, while 5-FU has five hydrogen bonds with ARG233, TYR230, GLN184, ARG87, and HIS144. Cardamonin demonstrated Pi-Pi T shaped bond via PHE221 and sigma bond with PHE219 (Appendix A). Moreover, it showed few alkyl bonds with MET294, VAL226, and ILE159 with aromatic groups of CADMN, which made the active pocket more stable and caused lower binding energy than the one in 5-FU. Free binding energy of the interaction between CADMN and caspase 7 showed to be −6.89 kcal/mol, while it was −4.68 kcal/mol for 5-FU.

Appendix A shows four hydrogen bindings which occurred after interaction between CADMN and caspase 8 via ARG260, GLN358, SER316, and GLY318. It also showed sigma binding with ILE257 and alkyl binding via CYS360 and aromatic group in the structure of CADMN. 5-FU showed six conventional hydrogen bonds with SER316, GLN358, ARG260, GLY318, CYS360, and ARG413. However, free binding energy of such interaction between CADMN and caspase 8 (−7.14 kcal/mol) was much lower than the one in 5-FU (−5.66 kcal/mol).

Cardamonin interactions with caspase 9 have shown five conventional hydrogen bonds with ARG372, VAL379, ASN375, ASP150, and ASN148. It has shown pi-sigma bond with ALA376 and Pi-alkyl binding with LYS380, ILE382, and LEU151. However, 5-FU demonstrated four hydrogen bonds with GLY388, TYR331, THR333, and GLN240 (Appendix A). Based on the molecular docking results, CADMN showed the free binding energy of −8.75 kcal/mol, while 5-FU, which was chosen as positive control, showed −5.01 kcal/mol. The lower free binding energy of CADMN compared with 5-FU in interaction with caspase 9 could be due to the higher number of hydrogen bonds that were present and also due to the stability of the active pocket where alkyl binding occurred between aromatic group of CADMN and LYS380, ILE382, and LEU151 of caspase 9.

Appendix A shows CADMN demonstrated three hydrogen bonds with PCNA including GLU124, LEU126, and GLN38. There is also pi-sigma bond with SER39 and Pi-alkyl bond with ILE23. 5-FU showed four hydrogen bonds with PCNA protein but with lower free binding energy of −4.83 kcal/mol than the one in CADMN (−8.01 kcal/mol). This is probably due to higher stability of active pocket occurred in PCNA because of pi-sigma binding with the aromatic group of CADMN.

Appendix A shows CADMN demonstrated three hydrogen bonds with NF-κB via GLU25, SER56 and TRP33. Moreover, it showed alkyl binding with ALA54 and PRO55. 5-FU demonstrated same numbers of hydrogen bonds via SER6, LYS10 and MET9. However, it showed free binding energy of −3.74 kcal/mol, while CADMN showed −5.53 kcal/mol. This can be due to extra stability present in the active pocket of NF-κB because of the structure of CADMN involved in the interaction.

Appendix A shows that the interaction between CADMN and NF-κB-p65 demonstrated three hydrogen bonds with ILE196, LEU194, and THR191. It also showed pi-sigma bond with MET284 and Pi-alkyl bond with LYS195. The positive control 5-FU demonstrated four hydrogen bindings via GLY237, VAL226, ILE224, and PHE239. Free binding energy of CADMN after interaction with NF-κB-p65 showed to be −7.28 kcal/mol, while it was −4.43 kcal/mol for 5-FU. Even though the number of hydrogen bonds in 5-FU is more than CADMN, the stability of the binding pocket in NF-κB-p65 by interacting with CADMN was more than 5-FU. Therefore, it caused higher free binding energy and lower affinity towards 5-FU compared with CADMN.

Appendix A shows CADMN has only one hydrogen binding with cytochrome C via TYR123, while 5-FU showed three hydrogen bonds via SER52, LYS49, and GLN87. Cardamonin free binding energy of −7.45 kcal/mol has shown more than 2 times more affinity to interact with cytochrome C compared with 5-FU (−3.72 kcal/mol). Cardamonin showed some alkyl bindings with CYS116, LEU16, ALA68, and ARG124 as well as pi-sulfur binding with CYS119. This made the active pocket of cytochrome C more stable than the time interacting with 5-FU. This contributed to CADMN having lower free binding energy than 5-FU. A lower concentration was therefore needed to inhibit cytochrome C.

## 4. Discussion

Safety profiles for bioactive compounds with herbal origin should be established through extensive preclinical toxicity tests [26]. In this study, acute toxicity test was conducted using animal model, and the results showed that the administration of either 300 or 2000 mg /kg of CADMN did not result in clinical toxic signs or death in female ICR mice. In addition, evidence from liver and kidney histological sections showed normal architecture indicating that cardamonin was safe and well-tolerated up to 2000 mg/kg. Similarly, previous sub-chronic toxicity study demonstrated no mortality has been reported after treatment with CADMN in rats at 5 mg/kg (i.p.) dose for 28 days. Throughout the observation period, the rats did not show any signs of toxicity or abnormal behaviors [27] emphasizing that CADMN has low toxicity profile in animals.

CADMN was reported to be a potential and promising anticancer chalcone [28]. The antitumor effects of CADMN in HepG2 cell xenograft nude mice have been demonstrated in vivo, while the in vitro cytotoxicity data indicated that CADMN exerts dose-dependent inhibition against HepG2 cells. The findings of this study established that CADMN efficiently inhibited tumor growth in the HepG2 cells xenograft nude mice, while there were no toxic effects on both liver and kidney functions. Similarly, in previous study, breast cancer was induced in Balb/c mice by injecting breast cancer cells (4T1) into the mammary fat pad of mice. Upon establishing breast cancer, 2.5 and 5 mg/kg of cardamonin were injected intraperitoneally every two days, which resulted in a significant decreased in tumor volume [12]. The results suggested that CADMN caused remarkable changes in tumor tissues which is also in agreement with previous study of CADMN on colorectal cancer exhibiting severe inflammation associated with dysplasia in CADMN-treated group [29].

Cell proliferation is an essential process during carcinogenicity; therefore, proliferating cell nuclear antigen (PCNA) was assessed since it is highly conserved nuclear protein of DNA polymerase-delta, and up-regulation the expression of PCNA and Ki-67 indicates proliferative activity of tumor cells [30]. Accordingly, the treatment with CADMN showed down-regulation in the expression of PCNA indicating an anti-proliferative effect, which was consistent with the findings of the proliferation suppression effects of CADMN on different cancer cells [31,32,33].

Triggering apoptosis signaling pathway in cancer cells is considered one of the strategies to explore the molecular mechanisms of anti-cancer agents in drug discovery. Cellular apoptosis can be initiated through the internal or external pathways under the effect of these agents.

The findings of our study indicated that CADMN has anti-tumorigenic effect on HepG2 cells as a result of apoptosis and inhibition of NF-κB pathway via down regulation of anti-apoptotic Bcl-2 protein, p-NF-κB-p65, and Ikkβ proteins, while pre-apoptotic Bax protein was significantly over-expressed.

It has been reported that CADMN suppresses the signaling of NF-κB pathway, which was manifested to induce the process of the apoptosis [34,35]. Similarly, our in vitro results showed that NF-κB translocation from cytoplasm to the nucleus was significantly inhibited in CADMN-treated HepG2 cells as compared to untreated cells, which suggests the involvement of NF-κB in the anti-tumorigenic action of CADMN [15]. The in vivo findings support this suggestion based on the results of the immunofluorescence, which showed significant lower expression level of p-NF-κB-p65 upon CADMN treatment.

## 5. Conclusions

The present study demonstrated that cardamonin has an anti-tumorigenic effect against HCC, which showed significant reduction in the tumor volume and tumor growth rate in treated nude mice. Acute toxicity study demonstrated that CADMN is safe and well tolerated up to 2000 mg/kg. Our results indicated that CADMN induces apoptosis and promotes the activation of Bax and inhibits Bcl-2 and p-NF-κB-p65 proteins expression.

The docked conformation of Bax BH3, Bcl2, Cyt C, caspase 7, caspase 8, caspase 9, PCNA, NF-κB, and NF-κB-p65 with active conformation of each ligand consists of CADMN and 5-FU clearly revealed that numerous potential interactions were present. These findings suggest that the anti-tumorigenic effect of CADMN in HepG2 cells may be mediated via NF-κB pathway. CADMN could therefore be considered as a potential treatment for liver cancer; however, further study should extensively be carried out to shed light to more notable findings.

## Figures and Tables

**Figure 1 biomedicines-08-00586-f001:**
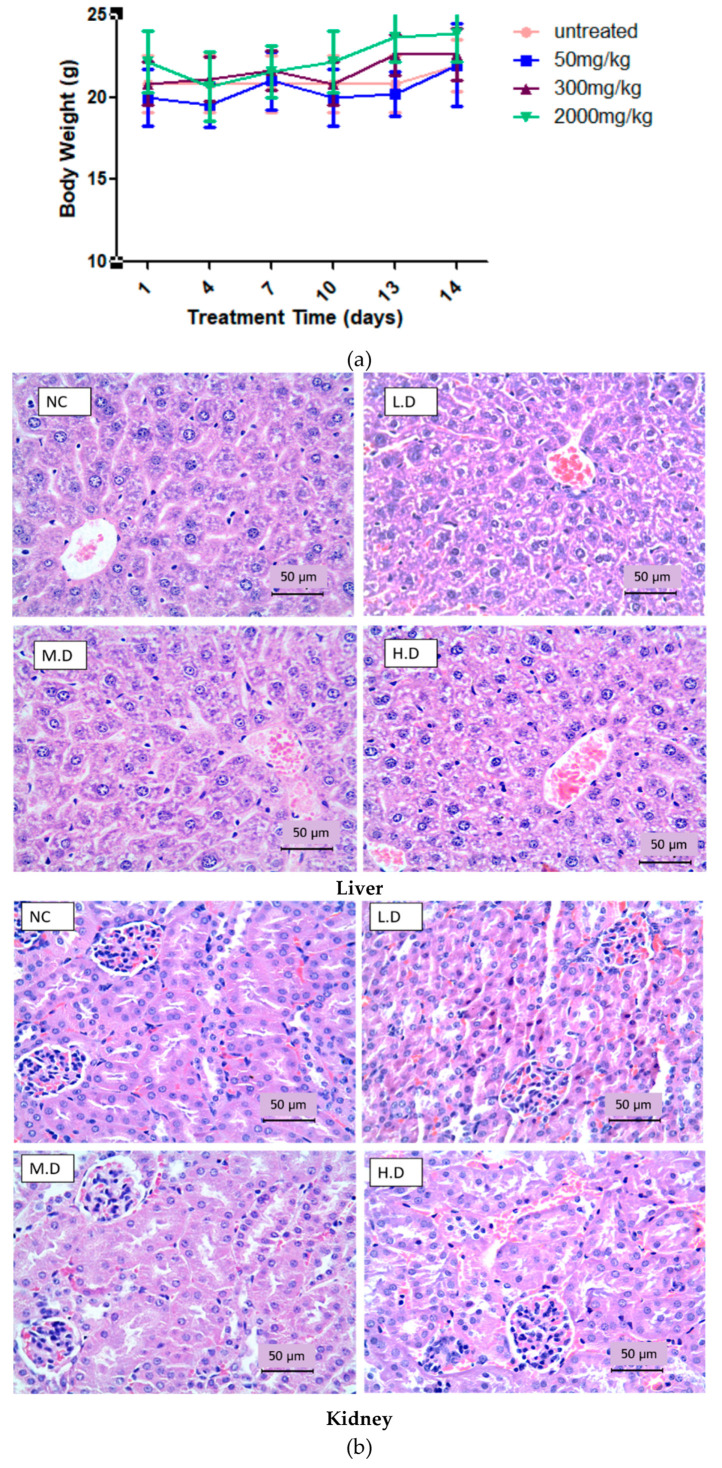
Acute toxicity study of cardamonin (**a**) Body weight changes of CADMN-treated groups and untreated group for 14 days in female ICR mice. Data were expressed as mean ± standard deviation. (**b**) Histology sections of liver and kidney in control and CADMN-treated ICR mice. Animals were observed after a single oral dose for 14 days. Hematoxylin and Eosin staining were applied on liver and kidney tissues after harvesting. Toxicity signs were absence in the liver and kidney histological samples. Magnification 40×, NC: negative control, LD: low dose (50 mg/kg), MD: moderate dose (300 mg/kg) and HD: high dose (2000 mg/kg).

**Figure 2 biomedicines-08-00586-f002:**
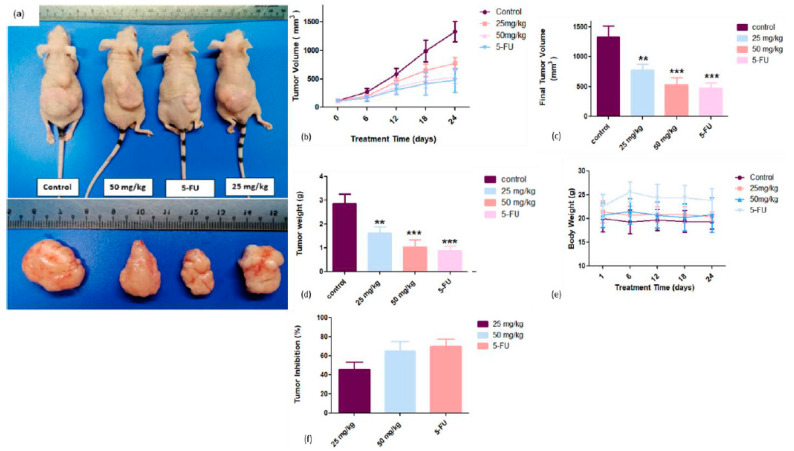
Tumor size, tumor volume, final tumor volume, tumor weight, body weight, and tumor inhibition rate of treated and untreated nude mice. (**a**) Photographs represent tumor size of CADMN-treated and untreated mice. (**b**) Tumor volume of CADMN-treated and control untreated groups throughout the treatment for 24 days. (**c**) Mean final tumor volume in mm^3^ of CADMN-treated and untreated groups. (**d**) Mean tumor weight in gram of CADMN-treated and untreated mice after 24 days of treatment. (**e**) Body weight changes in CADMN-treated and untreated mice throughout the treatment period. (**f**) Tumor volume inhibitory rates of CADMN-treated and positive control group (5-FU). Results were expressed as mean ± standard deviation. ** *p* < 0.01, *** *p* < 0.001 indicate significant differences as compared to control.

**Figure 3 biomedicines-08-00586-f003:**
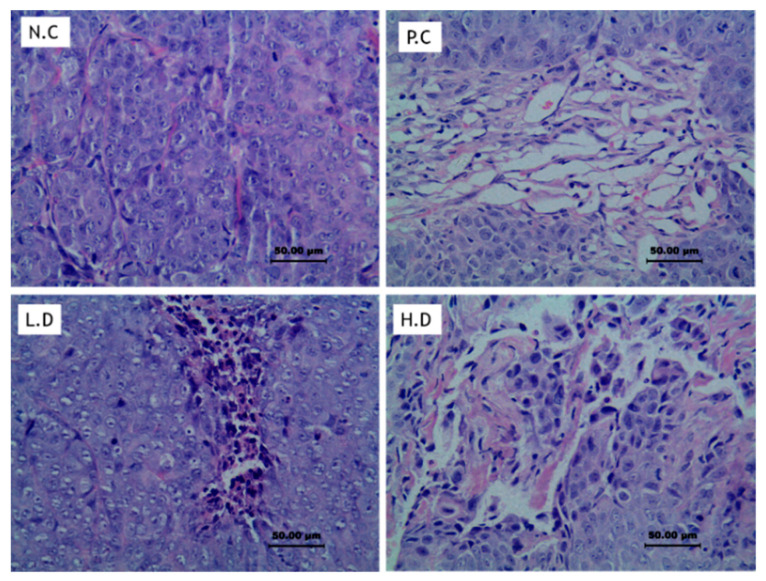
Histology sections of control and CADMN-treated nude mice. The sections of untreated group (NC) showed large solid tumor tissues, sections treated mice (H.D and P.C) showed larger area of tumor necrotic characterized by cell shrinkage, fragmentation, and chromatin disappearance with a small island of tumor tissues. Magnification 40 ×, NC: negative control, PC: positive control (5-FU), LD: low dose (25 mg/kg CADMN), and HD: high dose (50 mg/kg CADMN).

**Figure 4 biomedicines-08-00586-f004:**
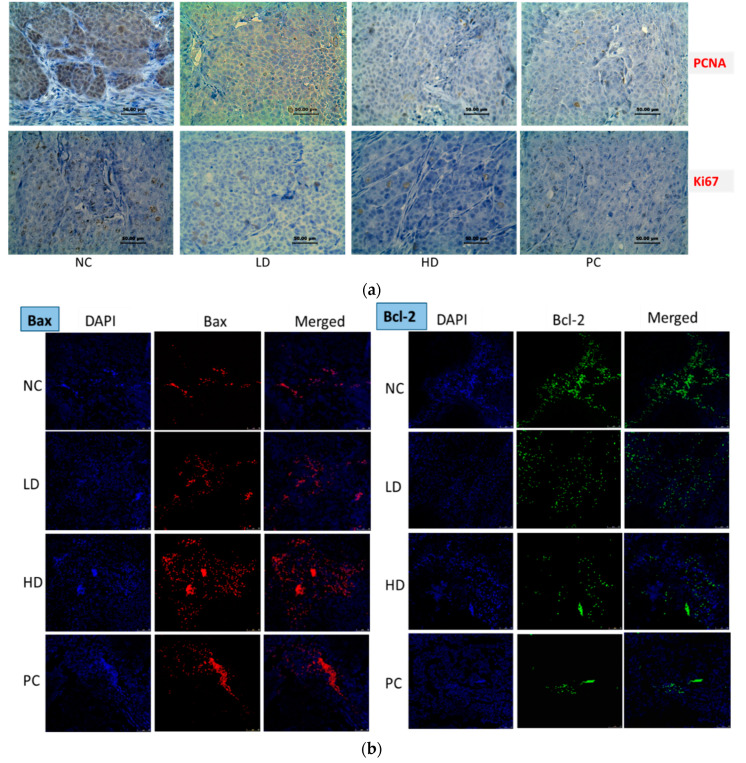
Immunohistochemistry and immunofluorescence of CADMN-treated and untreated nude mice (**a**) Immunohistochemistry staining of Ki-67 and PCNA proteins for each group, (**b**) Immunofluorescence signals of Bax and Bcl-2, magnification 20x and scale bars 100 μm. (**c**) Immunofluorescence signals of p-NF-κB-p65 and IKKβ expression in tumor tissues. Red signals indicate Bax and p-NF-κB-p65 expression and green signals indicate Bcl-2 and Ikkβ expression, magnification 20x and scale bars 100 μm. (**d**) Quantitative analyses of fluorescence signal intensity in tumor tissues for each protein. NC: negative control, LD: low dose (25 mg/kg CADMN), HD: high dose (50 mg/kg CADMN) and PC: positive control (5-FU). Each value is expressed as mean ± SEM. (n = 6). * *p* <0.05, ** *p* < 0.01, *** *p* <0.001 indicate significant differences between groups, ns indicates no significant differences between groups.

**Table 1 biomedicines-08-00586-t001:** Biochemical parameters analysis for liver function (ALT, ALP, AST, and T-bilirubin) and kidney function (creatinine and urea) parameters.

Parameters	Negative Control	25 mg/kg CADMN	50 mg/kg CADMN	Positive Control 30 mg/kg 5-FU
ALT (U/L)	88 ± 40.3	60 ± 28.0	55 ± 21.0	272 ± 107.0 *
ALP (U/L)	460 ± 208.1	479 ± 194.5	425 ± 160.3	539 ± 137.4
AST (U/L)	89 ± 39.2	77 ± 13.0	86 ± 45.3	90 ± 17.7
T-BIL (μmol/L)	2 ± 0.14	2 ± 0.3	2 ± 28.0	3 ± 0.7
UREA (mmol/L)	8 ± 1.4	8 ± 1.4	10 ± 2.3	9 ± 1.5
Cr (μmol/L)	33 ± 4.0	33 ± 3.6	36 ± 14.0	34 ± 11.5

* *p* < 0.05 indicates significant differences as compared with negative control.

**Table 2 biomedicines-08-00586-t002:** Free binding energy of cardamonin and 5-FU after interaction with different proteins.

**CADMN**										
	Cyt. C	Bax	Caspase 3	Caspase 7	Caspase 8	Caspase 9	PCNA	NF-κB	NF-κB-p65	Bcl-2
Run	51	51	37	14	60	71	98	11	6	58
FBE (kcal/mol)	−7.45	−6.11	−7.16	−6.89	−7.14	−8.75	−8.01	−5.53	−7.28	−6.27
Ki	3.45 µM	33.23 µM	5.69 µM	8.97 µM	5.84 µM	387.52 nM	1.35 µM	88.49 µM	4.57 µM	25.50 µM
**5-FU**										
	Cyt. C	Bax	Caspase 3	Caspase 7	Caspase 8	Caspase 9	PCNA	NF-κB	NF-κB -p65	Bcl-2
Run	99	12	94	93	72	35	43	39	73	79
FBE (kcal/mol)	−3.72	−3.59	−5.40	−4.68	−5.66	−5.01	−4.83	−3.74	−4.43	−4.03
Ki	1.87 mM	2.32 mM	110.66 µM	368.75 µM	70.78 µM	211.39 µM	288.20 µM	1.83 mM	567.87 µM	1.11 mM

FBE: free binding energy, Ki: inhibition constant.

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
