# Peer review of "Cardamonin Exerts Antitumor Effect on Human Hepatocellular Carcinoma Xenografts in Athymic Nude Mice through Inhibiting NF-κβ Pathway"

_biomedicines, 2020, doi:10.3390/biomedicines8120586_

Round 1

Reviewer 1 Report

Overall comments:

  1. The work by Badroon et al. titled ‘Cardamonin Exerts Antitumor Effect on Human Hepatocellular Carcinoma Xenografts in Athymic Nude Mice through Inhibiting NF-κβ Pathway’ is a very significant work and can be considered for publication only after careful revision.
  2. All the figure legends can be clarified in more details as they are independent. I think, this will increase the value of the paper and will be easy to understand for the readers.
  3. Extensive English editing is required.
  4. Some of the main figures, may be 4 to 6 figure could have been in the main text and rest of the supporting figures could have been presented in supporting documents. The text of the figures (somewhere) are not clear for font size.
  5. The references have not been cited properly. Please verify all the cited references throughout the manuscript.

Specific comments:

Page 1 line 21: Since previous studies showed that should be deleted.

Page 12 figure 6: has the figure drawn by the authors or has taken from somewhere. If it is taken from somewhere, please acknowledge properly.

Figure 2: the scale bars are not clear. Please increase the font size and it can be presented as only 50 μm (50.00 μm).

Page 2 line 45: In addition, the incidence rate of HCC 45 has been reported to vary considerably across the world, with the highest incidence observed in East Asia. On the other hand, several problems have been arisen in the treatment of HCC including drug resistance and high recurrence rate…the reference has been cited wrongly. Please put the right reference for this information.

Page 2 line 56-58: In animals, CADMN was also able to show inhibitory effects on tumor growth in mice through modulation of different pathways such as STAT3, Wnt/β-catenin, mTOR, and NF- κB signaling pathways [7-10]…number 7 reference is on cell line not in animal model.

Author Response

No.

Reviewer’s Comment

Authors’ Response

1

All the figure legends can be clarified in more details as they are independent. I think, this will increase the value of the paper and will be easy to understand for the readers.

Thank you for your comment. The figure legends has been improved to enabling the figures to stand alone.

2.

Extensive English editing is required.

English editing has been done thoroughly and the amendments are highlighted in red.

3.

Some of the main figures, may be 4 to 6 figure could have been in the main text and rest of the supporting figures could have been presented in supporting documents. The text of the figures (somewhere) are not clear for font size.

Figures 1 and 6-15 (docking data) have been moved to supplementary materials. Only 4 figures are in the main text, while the rest are uploaded as supplementary i.e., Figures S1-S11.

4.

The references have not been cited properly. Please verify all the cited references throughout the manuscript.

All the cited references were checked and corrected accordingly. In addition, four more references were cited and added to the reference list.

5.

Page 1 line 21: Since previous studies showed that should be deleted.

The sentence was deleted as advised (line 23).

6.

Page 12 figure 6: has the figure drawn by the authors or has taken from somewhere. If it is taken from somewhere, please acknowledge properly.

Figure 6 has been drawn using the mentioned software. (Line 185-187 & 190-192).

7.

Figure 2: the scale bars are not clear. Please increase the font size and it can be presented as only 50 μm (50.00 μm).

The scale bars were improved to be made clearer and the decimal point was deleted.

8.

Page 2 line 45: In addition, the incidence rate of HCC 45 has been reported to vary considerably across the world, with the highest incidence observed in East Asia. On the other hand, several problems have been arisen in the treatment of HCC including drug resistance and high recurrence rate…the reference has been cited wrongly. Please put the right reference for this information.

The right references were cited for both sentences (references 2 and 3, page 2).

9.

Page 2 line 56-58: In animals, CADMN was also able to show inhibitory effects on tumor growth in mice through modulation of different pathways such as STAT3, Wnt/β-catenin, mTOR, and NF- κB signaling pathways [7-10]…number 7 reference is on cell line not in animal model.

Appropriate and right references were cited (references 10, 12-14, page 2).

Reviewer 2 Report

I strongly recommend the acceptance of the manuscript in the present form.

The manuscript titled: Cardamonin Exerts Antitumor Effect on Human Hepatocellular Carcinoma Xenografts in Athymic Nude Mice through Inhibiting NF-κβ Pathway and authorized by Badroon et al. deserves to be considered for publication as it included an interesting and new work ideas with very good sound effect in the research area and consequently, it will be interesting for the readers.

Author Response

Reviewer’s Comment

Authors’ Response

1

The manuscript titled: Cardamonin Exerts Antitumor Effect on Human Hepatocellular Carcinoma Xenografts in Athymic Nude Mice through Inhibiting NF-κβ Pathway and authorized by Badroon et al. deserves to be considered for publication as it included an interesting and new work ideas with very good sound effect in the research area and consequently, it will be interesting for the readers.

Thank you for the encouraging and positive feedback.